# Cyanidin Increases the Expression of Mg^2+^ Transport Carriers Mediated by the Activation of PPARα in Colonic Epithelial MCE301 Cells

**DOI:** 10.3390/nu11030641

**Published:** 2019-03-16

**Authors:** Yui Takashina, Aya Manabe, Yoshiaki Tabuchi, Akira Ikari

**Affiliations:** 1Laboratory of Biochemistry, Department of Biopharmaceutical Sciences, Gifu Pharmaceutical University, Gifu 501-1196, Japan; 145037@gifu-pu.ac.jp (Y.T.); 135072@gifu-pu.ac.jp (A.M.); 2Life Science Research Center, University of Toyama, Toyama 930-0194, Japan; ytabu@cts.u-toyama.ac.jp

**Keywords:** CNNM4, cyanidin, Mg^2+^ deficiency, PPAR, TRPM6

## Abstract

Mg^2+^ deficiency may be involved in lifestyle-related diseases, including hypertension, cardiovascular diseases, and diabetes mellitus. Dietary Mg^2+^ is absorbed in the intestine mediated through transcellular and paracellular pathways. However, there is little research into what factors upregulate Mg^2+^ absorption. We searched for food constituents that can increase the expression levels of Mg^2+^ transport carriers using mouse colonic epithelial MCE301 cells. Cyanidin, an anthocyanidin found in black beans and berries, increased the mRNA levels of Mg^2+^ transport carriers including transient receptor potential melastatin 6 (TRPM6) channel and cyclin M4 (CNNM4). The cyanidin-induced elevation of Mg^2+^ transport carriers was blocked by GW6471, a peroxisome proliferator-activated receptor α (PPARα) inhibitor, but not by PPARγ, PPARδ, and protein kinase A inhibitors. Cyanidin-3-glucoside showed similar results to cyanidin. Cyanidin increased the protein levels of TRPM6 and CNNM4, which were distributed in the apical and lateral membranes, respectively. The nuclear localization of PPARα and reporter activities of Mg^2+^ transport carriers were increased by cyanidin, which were inhibited by GW6471. The cyanidin-induced elevation of reporter activity was suppressed by a mutation in a PPAR-response element. Fluorescence measurements using KMG-20, an Mg^2+^ indicator, showed that Mg^2+^ influx and efflux from the cells were enhanced by cyanidin, and which were inhibited by GW6471. Furthermore, cyanidin increased paracellular Mg^2+^ flux without affecting transepithelial electrical resistance. We suggest that cyanidin increases intestinal Mg^2+^ absorption mediated by the elevation of TRPM6 and CNNM4 expression, and may constitute a phytochemical that can improve Mg^2+^ deficiency.

## 1. Introduction

Magnesium ions (Mg^2+^) play an important role in the regulation of various physiological functions. Over 300 enzymes, including those involved with oxidative phosphorylation and glycolysis, require Mg^2+^ for their activity. Chronic Mg^2+^ deficiency may be one of the causes of lifestyle-related diseases such as hypertension, cardiovascular diseases, and diabetes mellitus [1,2,3]. In addition, Mg^2+^ is necessary to protect individuals from physical and mental stresses [4,5]. Mg^2+^ deficiency due to reduced dietary intake in healthy people may be rare. However, many people are exposed to excessive stress in their social environment, and such stress is thought to be increasing and limiting to health complications. Mg^2+^ has the capability to relieve tension stress. Therefore, Mg^2+^ consumption and excretion are growing at an impressive rate. Moreover, high fat and energy diets may lead to defects in Mg^2+^ intake. Individuals under excessive stress and with an unbalanced diet may have a higher risk of chronic Mg^2+^ deficiency. Excess intake of Mg^2+^ may cause diarrhea, resulting in the further deterioration of Mg^2+^ deficiency. Therefore, it may be more effective to take a food constituent that enhances Mg^2+^ absorption.

The daily intake of Mg^2+^ for a normal adult is about 300–400 mg and approximately 30%–50% is absorbed in the intestine. The intestinal absorption rate of Mg^2+^ depends on the amount of Mg^2+^ intake [6] and is enhanced when Mg^2+^ intake is low [7]. The absorption of Mg^2+^ occurs mostly in the small intestine and colon, and the absorption routes are divided into two pathways, transcellular, and paracellular. The absorption of Mg^2+^ in the colon mainly depends on the transcellular pathway [8]. The transient receptor potential melastatin 6 (TRPM6) Mg^2+^ channel is exclusively expressed at the apical membrane of the colon and renal distal convoluted tubules [9]. The *TRPM6* gene has been identified as the causative gene of a rare autosomal recessive disorder, hypomagnesemia with secondary hypocalcemia [10,11]. On the other hand, TRPM7, a close homologue of TRPM6, is expressed ubiquitously in a broad spectrum of tissues. Cyclin M4 (CNNM4, previously known as ancient domain protein 4) is expressed at the basolateral membrane of the intestine and is considered to function as an Mg^2+^ transporter [12]. A correlation between reduced serum Mg^2+^ concentration and single nucleotide polymorphisms in *CNNM* genes, including CNNM4, has been shown by genome-wide association studies [13]. Both TRPM6 and CNNM4 may be responsible for the absorption of Mg^2+^ in the colon. However, the regulatory mechanism of expression of these Mg^2+^ transport carriers remains unclear.

Mg^2+^ absorption is impaired by some food components such as phytic acid and oxalate [14]. In contrast, magnesiotropic hormones including parathyroid hormone, 1,25 dihydroxyvitamin D, and epidermal growth factor (EGF) have been suggested to up-regulate Mg^2+^ absorption [15], but there are few reports about food components. Mg^2+^ is abundant in nuts, vegetables, and fruits [16]. Black soybean, which contains various bioactive components, including flavonols, anthocyanins, polyphenols, and linoleic acid, have been reported to be useful in supplying Mg^2+^ in serum [17]. Cyanidin-3-O-glucoside (cyanidin-3G), one of the most common anthocyanins, is absorbed in the intestine and may produce cyanidin through hydrolysis by β-glucosidase [18]. Both cyanidin and cyanidin-3G have various bioactivities, such as anti-tumor, anti-infection, and anti-diabetic properties [19], but there are no reports concerning mineral absorption in the colon.

In the present study, we found that cyanidin increases the mRNA and protein levels of TRPM6 and CNNM4 in mouse colonic epithelial MCE301 cells. Consequently, the effects of cyanidin on intracellular localization, transcriptional activity, and Mg^2+^ flux through TRPM6 and CNNM4 were investigated using immunofluorescence measurements, luciferase assay, and Mg^2+^ fluorescence measurements, respectively. In addition, the binding of peroxisome proliferator-activated receptor α (PPARα) to the promoter region of these Mg^2+^ transport carriers was determined using chromatin immunoprecipitation (ChIP) assays. Our findings indicate that cyanidin may be useful to increase Mg^2+^ absorption in the intestine and prevent chronic Mg^2+^ deficiency.

## 2. Experimental Section

### 2.1. Materials

Genistein, GW9662, and linoleic acid were purchased from Wako Pure Chemical Industries (Osaka, Japan). GSK3787, GW6471, and H-89 were from Cayman Chemical (Ann Arbor, MI, USA). Cyanidin and cyanidin-3G were from TOKIWA PHYTOCHEMICAL (Chiba, Tokyo) and FUJICCO (Kobe, Japan), respectively. Anti-CNNM4, anti-PPARα, anti-TRPM6 (CHAK2), and anti-TRPM7 antibodies were from GeneTex (Hsinchu, Taiwan), Rockland (Limerick, PA, USA), Abgent (San Diego, CA, USA), and Imgenex (San Diego, CA, USA), respectively. All other reagents were of the highest grade of purity available. The pharmacological effects of drugs are listed in Table 1.

### 2.2. Cell Culture

Mouse colonic non-carcinoma epithelial MCE301 cells derived from transgenic mice [20] were grown in Dulbecco’s modified Eagle’s medium/Ham’s nutrient mixture F-12, as described previously [21]. The cells were cultured in the medium without fetal bovine serum the day before the experiments. Control cells were treated with dimethyl sulfoxide (DMSO) as a vehicle.

### 2.3. RNA Isolation and Quantitative Real-Time Reverse-Transcription Polymerase Chain Reaction (PCR)

Total RNA was isolated from cells using TRI reagent (Sigma-Aldrich, St. Louis, MO, USA). Reverse transcription and quantitative real-time PCR was performed, as described previously [21]. The primer pairs used for PCR are listed in Table 2. β-Actin was used for normalization.

### 2.4. Preparation of Cytoplasmic Extracts and Western Blotting

Cytoplasmic extracts, which include plasma membrane and cytosolic proteins, were prepared, as described previously [22]. The aliquots were applied to sodium dodecyl sulfate-polyacrylamide gel electrophoresis and blotted onto a polyvinylidene fluoride membrane. After blocking with 2% skim milk at room temperature for 30 min, the membrane was incubated with each primary antibody (1:1000 dilution) at 4 °C for 16 h, followed by a peroxidase-conjugated secondary antibody (1:3000 dilution) at room temperature for 1.5 h. Finally, the membranes were incubated in EzWestLumi plus (ATTO Corporation, Tokyo, Japan) and scanned using a C-DiGit Blot Scanner (LI-COR Biotechnology, Lincoln, NE, USA). Band density was quantified using ImageJ software (National Institute of Health software). β-Actin was used for normalization of cytoplasmic proteins.

### 2.5. Immunocytochemistry

Cells cultured on Transwell plates (0.4 μm pore size, Corning Incorporated, Corning, NY, USA) were fixed with 3.7% paraformaldehyde at room temperature, then permeabilized with 0.2% Triton X-100 for 15 min. After blocking with 4% Block Ace (Dainippon Sumitomo Pharma, Osaka, Japan) for 30 min, the cells were incubated with primary antibodies at 4 °C for 16 h. They were then incubated with Alexa Fluor 488 and 546-conjugated antibodies in the presence of 4′,6-diamidino-2-phenylindole (DAPI) for 1.5 h at room temperature. Immunolabelled cells were visualized using an LSM700 confocal microscope (Carl Zeiss, Germany). The fluorescence intensity of PPARα was quantified using ImageJ software and the nuclear localization was shown as a percentage of total intensity.

### 2.6. Luciferase Reporter Assay

pGL4 luciferase reporter vectors (Promega, Madison, WI, USA) containing the promoter region of the *TRPM6* (NM_017662.4) or *CNNM4* (NM_033570.2) were constructed. A *Renilla* construct, pRL-TK vector (Promega), was used for normalizing transfection efficiency. Cells were transfected with plasmid vector using HilyMax (Dojindo Laboratories, Kumamoto, Japan). After 48 h of transfection, luciferase activity was assessed using the Dual-Glo Luciferase Assay System (Promega). The luminescence of the *firefly* and *renilla* luciferase was measured using an AB-2270 Luminescencer Octa (Atto Corporation, Tokyo, Japan). Using computer analysis (TRANSFAC databases, Match), putative PPAR response elements (PPREs) were identified in the promoter region between −1214 and −718 of the *TRPM6* gene and between −341 and −323 of the *CNNM4* gene. Mutants of PPREs were generated using a KOD mutagenesis kit (Toyobo Life Science, Osaka, Japan). The primer pairs used for the introduction of mutation are listed in Table 3.

### 2.7. ChIP Assay

Cells were treated with 1% formaldehyde to crosslink the protein to DNA. Then, ChIP assays were performed using an EpiQuik Chromatin Immunoprecipitation kit (Epigentek, Farmingdale, NY, USA) as recommended by the manufacturer’s instructions. To co-immunoprecipitate the DNA, anti-PPARα antibody was used. The eluted DNA was amplified by semi-quantitative and quantitative real-time PCR using the primer pairs as shown in Table 4. To confirm usage of the same amounts of chromatin in immunoprecipitation between groups, input chromatin was also used. ChIP data are represented as relative values in the cells treated with cyanidin alone.

### 2.8. Mg^2+^ Transport Assay

The change in intracellular free Mg^2+^ concentration ([Mg^2+^]_i_) was determined using a Mg^2+^-sensitive fluorescent dye, KMG-20 AM. Cells cultured on 96-well plates were loaded with Hanks balanced salt solution (HBSS) containing 137 mM NaCl, 5.4 mM KCl, 4.2 mM NaHCO_3_, 3 mM Na_2_HPO_4_, 0.4 mM KH_2_PO_4_, 5 mM HEPES, 1 mM CaCl_2_, and 10 mM glucose supplemented with 2 μM KMG-20 at 37 °C for 30 min. The KMG-20-loaded cells were washed twice with dye-free HBSS and the fluorescence was measured every 20 s at 535 nm after excitation at 430 and 485 nm using a fluorescence reader (Infinite F200 Pro, Tecan, Mannedorf, Switzerland). In the Mg^2+^ influx assay, MgCl_2_ (final concentration 1 mM) was added to the nominally Mg^2+^-free HBSS immediately after measurement start. [Mg^2+^]_i_ is represented as arbitrary units relative to a reference value measured at 0 min. The increases in fluorescence values for 5 min after measurement start (ΔA.U.) were compared in each group. In the Mg^2+^ efflux assay, the HBSS supplemented with 1 mM MgCl_2_ was replaced at the nominally Mg^2+^-free HBSS immediately after measurement started. The decreases in fluorescence values for 5 min after measurement start (ΔA.U.) were compared in each group. In the transepithelial Mg^2+^ flux assay, cells were cultured on Transwell plates. After forming the confluent monolayer, transepithelial electrical resistance (TER) was measured, as described previously [23]. The upper and lower chamber media were replaced to the nominally Mg^2+^-free HBSS. After 60 min of 1 mM MgCl_2_ addition to the upper chamber, the lower chamber solution was collected. The concentration of Mg^2+^ was measured by a colorimetric method using xylidyl blue-I (XB-I). XB-I formed a 520 nm absorbance maximum complex upon Mg^2+^ binding in alkaline conditions. After calibration, the concentration of Mg^2+^ was calculated.

### 2.9. Statistics

Results are presented as mean ± standard error of the mean. Differences between groups were analyzed using a one-way analysis of variance, and corrections for multiple comparison were made using Tukey’s multiple comparison tests. Comparisons between two groups were made using Student’s *t* test. Significant differences were assumed at *p* < 0.05.

## 3. Results

### 3.1. Increase in TRPM6 and CNNM4 mRNA Expression by Cyanidin

To identify food constituents that can increase intestinal Mg^2+^ absorption, we examined the effect of black bean components including genistein, cyanidin, and linoleic acid on the mRNA levels of Mg^2+^ transport carriers. Cyanidin significantly increased the mRNA levels of TRPM6 and CNNM4 without affecting that of TRPM7 (Figure 1). Similarly, cyanidin-3G increased the mRNA levels of TRPM6 and CNNM4 (Figure 2). Anthocyanin has been reported to activate protein kinase A (PKA) [24] and PPAR [25]. The cyanidin and cyanidin-3G-induced elevation of TRPM6 and CNNM4 mRNAs was significantly blocked by GW6471, a PPARα inhibitor. In contrast, GW9662, a PPARγ inhibitor, GSK3787, a PPARδ inhibitor, and H-89, a PKA inhibitor had no effect on the cyanidin-induced elevation of TRPM6 and CNNM4 mRNAs. These results indicate that cyanidin may increase the expression of both TRPM6 and CNNM4 mediated by the activation of PPARα.

### 3.2. Effect of Cyanidin on the Expression and Localization of TRPM6 and CNNM4 Proteins

TRPM6 may be localized in the apical membrane and play a role in the Mg^2+^ influx from the intestinal lumen to cells, whereas CNNM4 may be localized in the basolateral membrane and play a role in Mg^2+^ efflux from cells to the blood. Western blot analysis showed that the protein levels of TRPM6 and CNNM4 were increased by cyanidin, which were blocked by GW6471 (Figure 3A). These results coincided with those from real-time PCR. Immunofluorescence analysis showed that TRPM6 was localized to the apical membrane (Figure 3B). The fluorescence intensity of TRPM6 was enhanced by cyanidin, which was blocked by GW6471. In contrast, CNNM4 was detected in the nuclei and lateral membrane, but the expression was enhanced by cyanidin and was blocked by GW6471.

### 3.3. Increase in Nuclear Localization of PPARα by Cyanidin

PPARα was mainly localized in the cytosolic compartments in control conditions (Figure 4). Cyanidin increased the nuclear localization of PPARα. The effect of cyanidin was blocked by GW6471, but not by GW9662. These results supported the view that cyanidin may increase TRPM6 and CNNM4 expression mediated by the nuclear trafficking of PPARα.

### 3.4. Effects of Cyanidin and Inhibitors on TRPM6 and CNNM4 Promoter Activities

To clarify the effect of cyanidin on transcriptional activity, we measured the promoter luciferase activities of Mg^2+^ transport carriers. Cyanidin increased luciferase activities in cells transfected with wild-type constructs of TRPM6 or CNNM4 (Figure 5). The effects of cyanidin were significantly blocked by GW6471, but not by GW9662. The cyanidin-induced elevation of luciferase activities was also inhibited by the mutation of PPRE in the promoter region of TRPM6 and CNNM4. These results indicated that cyanidin may enhance the transcriptional activities of TRPM6 and CNNM4 mediated by the activation of PPARα.

### 3.5. Association of PPARα with the TRPM6 and CNNM4 Promoter Regions

In the ChIP assay, a primer pair that amplifies the region containing PPRE of TRPM6 and CNNM4 showed weak PCR signals in vehicle-treated cells (Figure 6). In contrast, PCR signals were detected in cyanidin-treated cells, which were blocked by GW6471. In the input images, similar PCR signals were detected in all samples. These results indicated that cyanidin may enhance the association of PPARα with the TRPM6 and CNNM4 promoter regions.

### 3.6. Increase in Mg^2+^ Transport by Cyanidin

To clarify whether TRPM6 and CNNM4 are functionally expressed in MCE301 cells, we examined the effect of cyanidin on Mg^2+^ transport. In the Mg^2+^ influx assay, the addition of 1 mM MgCl_2_ in the extracellular solution increased the fluorescence intensity of KMG-20, a Mg^2+^-sensitive fluorescent dye (Figure 7A). Cyanidin significantly enhanced the elevation of fluorescence intensity, which was blocked by GW6471. Similar results were observed using cyanidin-3G. Next, we investigated the effect of cyanidin on Mg^2+^ efflux. The removal of MgCl_2_ from the extracellular solution induced the decrease in the fluorescence intensity of KMG-20 (Figure 7B). Cyanidin significantly exaggerated the decrease in fluorescence intensity, which was blocked by GW6471. In contrast, GW9662 had no effects on cyanidin and cyanidin-3G-induced Mg^2+^ influx and efflux. These results indicated that cyanidin may increase the expression of functional TRPM6 and CNNM4. Finally, we examined the effect of cyanidin on transcellular ion permeability. TER was not significantly changed by the treatment with cyanidin, cyanidin-3G, or GW6471 (Figure 8). In contrast, transepithelial Mg^2+^ flux was enhanced by cyanidin and cyanidin-3G, which was completely inhibited by GW6471. These results indicated that cyanidin and cyanidin-3G can increase colonic Mg^2+^ transport.

## 4. Discussion

Dietary Mg^2+^ is absorbed through transcellular and paracellular pathways in the intestine. The paracellular transport of ions is restricted by claudins, tight junctional transmembrane proteins [26]. In contrast, it was for a long time unknown what carrier proteins are involved with transcellular Mg^2+^ absorption in the colon. Recently, the involvement of TRPM6 and CNNM4 was identified [9,12]. In the present study, we found that cyanidin and cyanidin-3G increased TRPM6 and CNNM4 expression in colonic MCE301 cells (Figure 1 and Figure 2).

Cyanidin-3G seems to be stable to acid pH and digestive enzymes in the stomach [27]. Approximately 1%–10% of cyanidin-3G is absorbed by the gastric epithelia mediated via transcellular pathways such as sodium-dependent glucose transporter (SGLT1), sodium-independent glucose transporter (GLUT1 and 3), and mono-carboxylated transporter 1. The transport system of cyanidin-3G in the intestine has been well characterized using Caco-2 cells derived from human colonic carcinoma. The absorption of cyanidin-3G is mainly regulated by SGLT1 and GLUT2 [28], but there might be another transport carrier for flavonols, because quercetin-3-glucose seems to inhibit cyanidin-3G absorption [29]. In contrast, cyanidin, the aglycon of cyanidin-3G, is more hydrophobic, thereby allowing diffusion across the plasma membrane. Both cyanidin-3G and cyanidin could increase the mRNA levels of TRPM6 and CNNM4 in MCE301 cells. Therefore, the structure of cyanidin may be necessary for the elevation of these Mg^2+^ transport carriers.

The cyanidin-induced elevation of TRPM6 and CNNM4 expression was inhibited by GW6471, but not by GW9662 and GSK3787, suggesting the involvement of PPARα. The nuclear trafficking of PPARα by cyanidin was blocked by GW6471, supporting the suggestion that cyanidin activates PPARα. Cyanidin binds directly to PPARα, PPARδ/β, and PPARγ, followed by activating the three PPAR subtypes-specific responsive genes [25]. The affinity for PPARα (K_D_: 3.08 μM) is higher than those for PPARγ (K_D_: 24.50 μM) and PPARδ/β (K_D_: 389.01 μM). Another reason may be explained by the difference in expression levels of PPAR subtypes. The expression level of PPARα is higher than those of PPARβ and PPARγ in the intestine of adult rats [30]. Therefore, the effect of cyanidin on Mg^2+^ transport carriers may mainly occur through the activation of PPARα.

The transcriptional mechanisms of *TRPM6* and *CNNM4* genes have not been clarified in detail. Thus far, we reported that EGF, tumor necrosis factor-α, and PPARγ agonists increased the mRNA levels of TRPM6 in renal tubular epithelial cells [22,31,32]. In contrast, there are no reports concerning the transcriptional regulatory factor of CNNM4. This is the first report showing that the transcriptional activities both of TRPM6 and CNNM4 are up-regulated by PPARα in colonic epithelial cells (Figure 5). The cyanidin-induced elevation of luciferase activity disappeared after treatment with GW6471 or mutation in a putative PPRE of the TRPM6 and CNNM4 promoters. Similarly, cyanidin increased the association of PPARα with the promoter region containing the PPRE, which was suppressed by GW6471. We suggest that cyanidin up-regulates the transcriptional activities of *TRPM6* and *CNNM4* genes mediated by the nuclear trafficking of PPARα and its binding to PPRE of the promoter region.

In the Mg^2+^ influx and efflux assays, cyanidin-3G and cyanidin significantly enhanced Mg^2+^ transport, which was blocked by GW6471 (Figure 7). These results suggested that both TRPM6 and CNNM4 are functionally expressed in MCE301 cells. Furthermore, cyanidin-3G and cyanidin increased transepithelial Mg^2+^ flux without affecting TER (Figure 8). We do not know the reason why TER was changed by cyanidin-3G and cyanidin in spite of the elevation of divalent cation flux. There are possibility that the flux of Mg^2+^ is negligible small compared to that of other ions, or cyanidin-3G and cyanidin can enhance the flux of anion. Anyway, we suggest that cyanidin-3G and cyanidin did not disrupt the paracellular barrier, but they can enhance Mg^2+^ absorption mediated through the transcellular pathway. The cell surface expression in the apical membrane of TRPM6 is regulated by Rac1, a small Rho GTPase [33]. On the other hand, the basolateral sorting of CNNM4 requires interactions with AP-1A and AP-1B [34]. Immunofluorescence analysis indicated that the green signal of TRPM6 is detected in the apical side (Figure 3B). In contrast, the red signal of CNNM4 was detected not only in the lateral membrane, but also in the nuclei. It may be useful to search for compounds that can enhance the trafficking of CNNM4 to the basolateral membrane.

## 5. Conclusions

In the present study, we found that cyanidin-3G and cyanidin increase TRPM6 and CNNM4 expression in MCE301 cells. Cyanidin-induced nuclear trafficking and association of PPARα with the promoter regions of TRPM6 and CNNM4 were significantly inhibited by GW6471. In addition, cyanidin increased reporter activity of TRPM6 and CNNM4, which was inhibited by mutation in PPRE sequences. Cyanidin-3G and cyanidin might increase TRPM6-mediated Mg^2+^ influx and CNNM4-mediated Mg^2+^ efflux, resulting in the elevation of transepithelial Mg^2+^ absorption in the colon. The presumed action mechanism of cyanidine-3G and cyanidin is illustrated in Figure 9. Foods abundant in cyanidin-3G might be sources of these phytochemicals that can improve Mg^2+^ deficiency.

## Figures and Tables

**Figure 1 nutrients-11-00641-f001:**
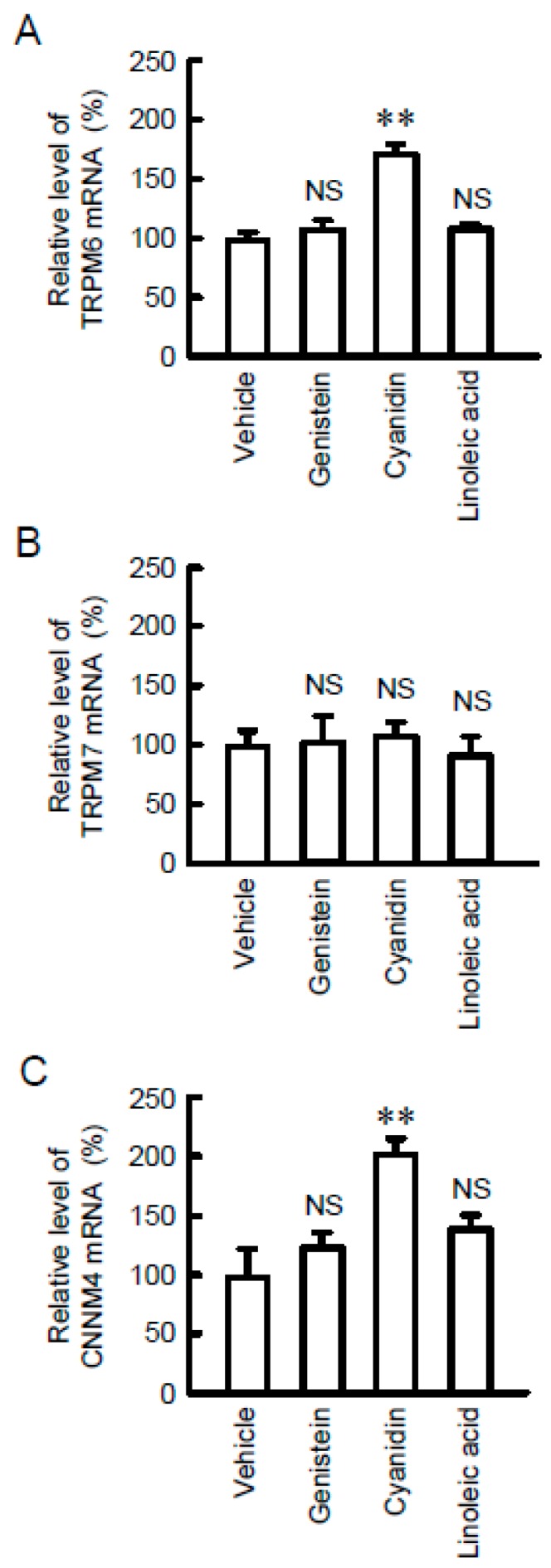
Increase in TRPM6 and CNNM4 expression levels by cyanidin treatment. MCE301 cells were incubated with vehicle (DMSO), 10 μM genistein, 10 μM cyanidin, or 10 μM linoleic acid for 6 h. After isolation of total RNA, quantitative real-time PCR was performed using primer pairs for TRPM6, TRPM7, CNNM4, and β-actin. The contents of TRPM6 (**A**), TRPM7 (**B**), and CNNM4 mRNAs (**C**) were represented relative to the values in vehicle. *n* = 4. ** *p* < 0.01 and NS *p* > 0.05 compared with vehicle.

**Figure 2 nutrients-11-00641-f002:**
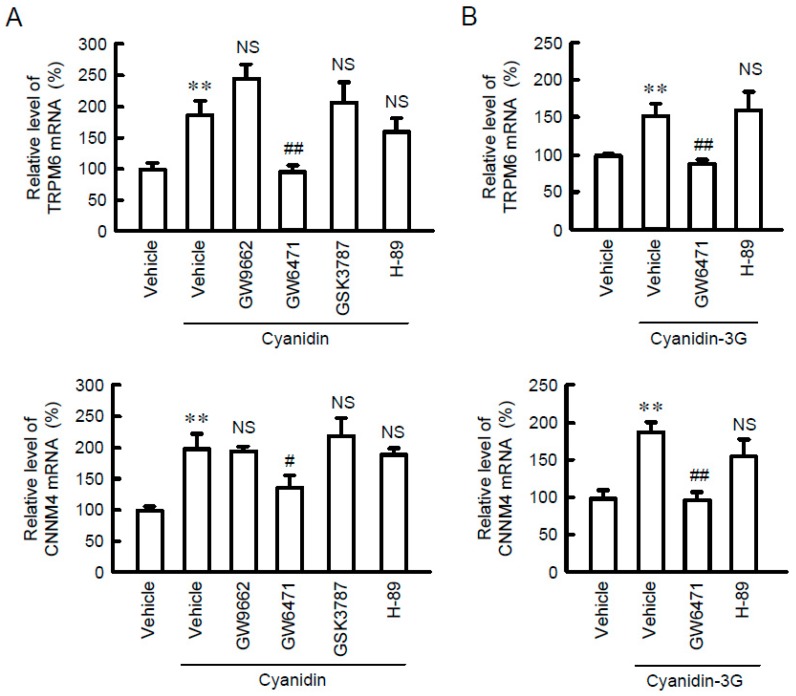
Inhibition of cyanidin-induced TRPM6 and CNNM4 expression by GW6471. (**A**) MCE301 cells were incubated with vehicle, 10 μM GW9662, 10 μM GW6471, 10 μM GSK3787, or 10 μM H-89 in the presence and absence of 10 μM cyanidin for 6 h. Quantitative real-time PCR was performed using primer pairs for TRPM6, CNNM4, and β-actin. The contents of TRPM6 and CNNM4 mRNAs are represented relative to the values in vehicle. (**B**) MCE301 cells were incubated with vehicle, 10 μM GW9662 or 10 μM H-89 in the presence and absence of 10 μM cyanidin-3G for 6 h. Quantitative real-time PCR was performed using primer pairs for TRPM6, CNNM4, and β-actin. The contents of TRPM6 and CNNM4 mRNAs are represented relative to the values in vehicle. *n* = 4. ** *p* < 0.01 and NS *p* > 0.05 compared with vehicle. ^##^
*p* < 0.01 and ^#^
*p* < 0.05 compared without cyanidin or cyanidin-3G alone.

**Figure 3 nutrients-11-00641-f003:**
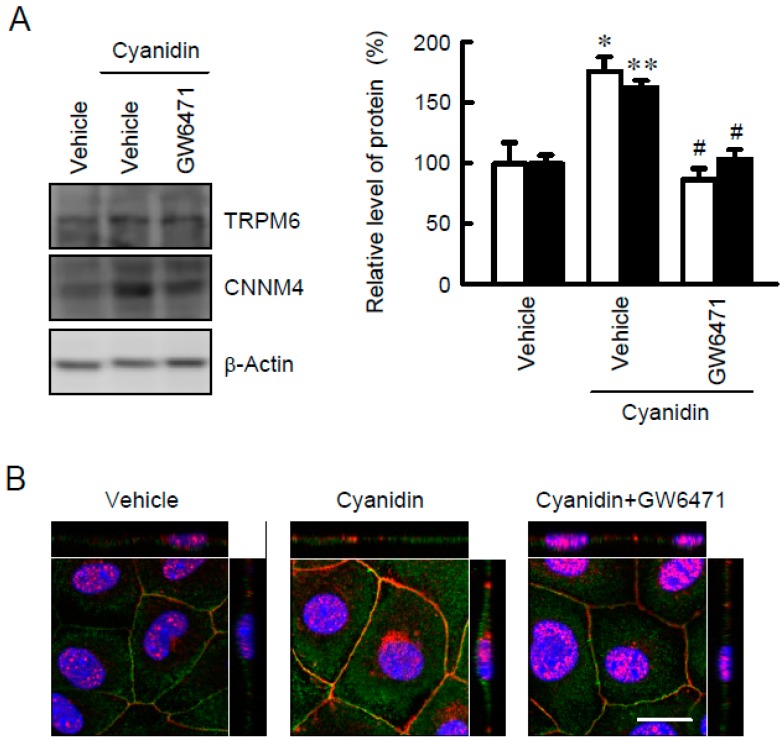
Increase in the expression levels of TRPM6 and CNNM4 protein by cyanidin treatment. (**A**) MCE301 cells were incubated with vehicle or 10 μM GW6471 in the presence and absence of 10 μM cyanidin for 24 h. Cytoplasmic extracts were immunoblotted with anti-TRPM6 or anti-CNNM4 antibodies. The expression levels were represented relative to the values with the absence of cyanidin. (**B**) MCE301 cells cultured on Transwell inserts were stained with anti-TRPM6 (green) and anti-CNNM4 (red) antibodies in the presence of DAPI (blue). The upper and right images represent the xz section. Scale bar indicates 20 μm. *n* = 4. ** *p* < 0.01 and * *p* < 0.05 compared without cyanidin. ^#^
*p* < 0.05 compared with cyanidin alone.

**Figure 4 nutrients-11-00641-f004:**
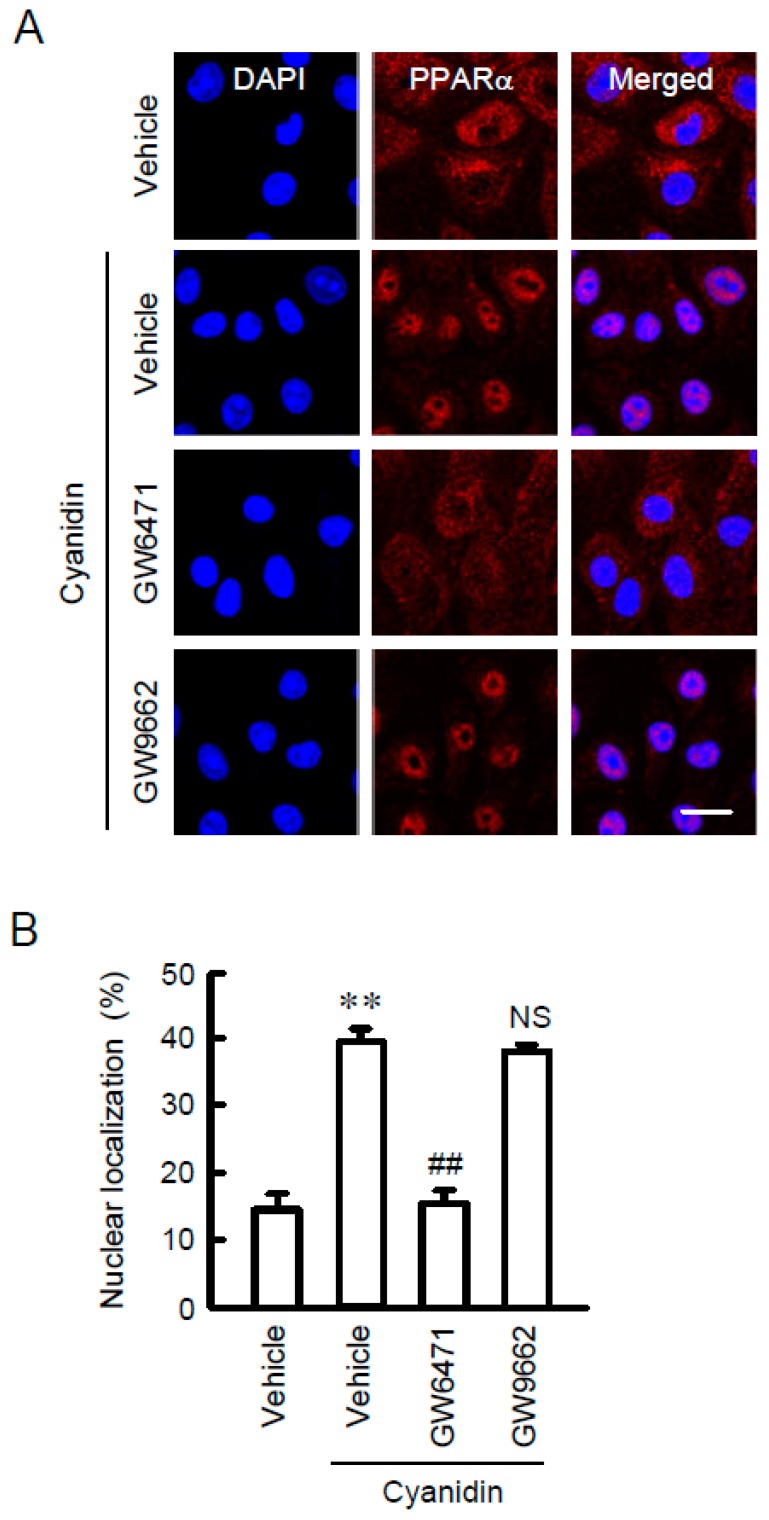
Increase in nuclear localization of PPARα by cyanidin. (**A**) MCE301 cells were incubated with vehicle, 10 μM GW6471, or 10 μM GW9662 in the presence and absence of 10 μM cyanidin for 1 h. The cells were stained with anti-PPARα antibody (red) and DAPI (blue). Merged images are shown in the right. Scale bar indicates 20 μm. (**B**) The fluorescence intensities of PPARα in the nuclei were measured using Image J and represented relative to the values of total fluorescence intensities. *n* = 4. ** *p* < 0.01 compared without cyanidin. ^##^
*p* < 0.01 and NS *p* > 0.05 compared with cyanidin alone.

**Figure 5 nutrients-11-00641-f005:**
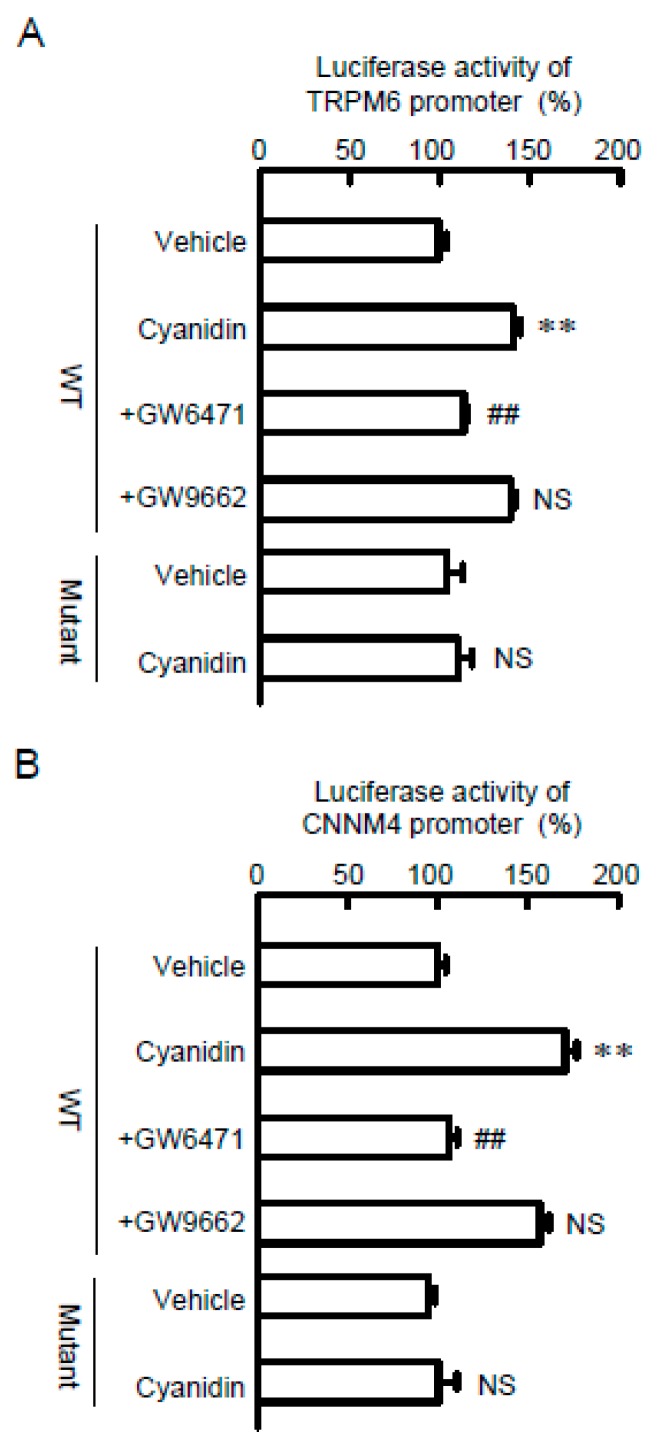
Involvement of PPARα on the cyanidin-induced elevation of TRPM6 and CNNM4 promoter activities. TRPM6 promoter (**A**) or CNNM4 promoter (**B**) luciferase vectors were co-transfected with a pRL-TK vector into MCE301 cells. WT and mutant indicate vectors with wild-type and mutant PPRE sequences, respectively. At 24 h after transfection, the cells were incubated with vehicle, 10 μM cyanidin, cyanidin plus 10 μM GW6471, or cyanidin plus 10 μM GW9662 for 6 h. The relative promoter activity was represented relative to the values in vehicle. *n* = 3–4. ** *p* < 0.01 and NS *p* > 0.05 compared with vehicle. ^##^
*p* < 0.01 compared without cyanidin alone.

**Figure 6 nutrients-11-00641-f006:**
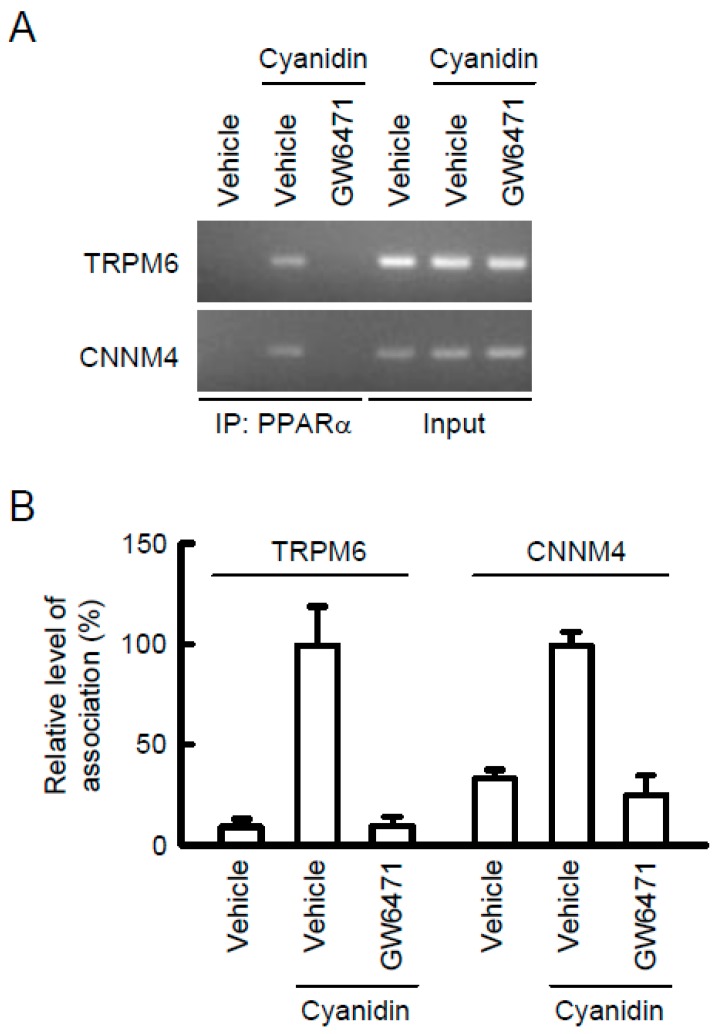
Association of PPARα with PPRE in the TRPM6 and CNNM4 promoters. MCE301 cells were incubated with vehicle or 10 μM GW6471 in the absence and presence of 10 μM cyanidin for 2 h. Genomic DNA was immunoprecipitated with an anti-PPARα antibody. After immunoprecipitation, the 5′-flanking region of TNRP6 was amplified by semi-quantitative (**A**) and quantitative PCR (**B**) using primer pairs amplifying PPRE from TRPM6 and CNNM4. Input was amplified by the primer without immunoprecipitation. ChIP data are represented relative to the values in cyanidin alone. *n* = 3.

**Figure 7 nutrients-11-00641-f007:**
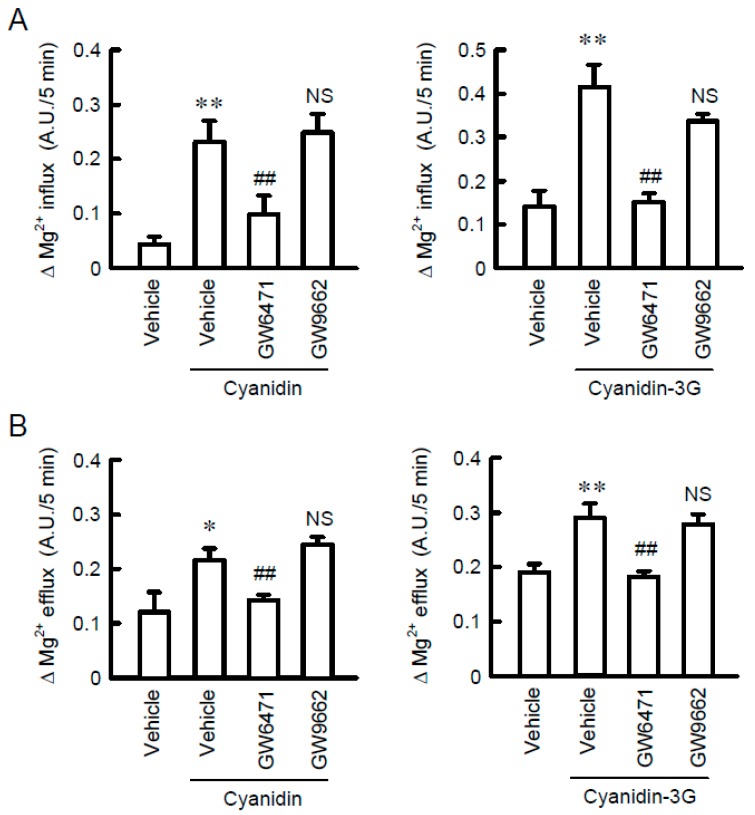
Elevation of Mg^2+^ influx and efflux from cells by cyanidin. (**A**) MCE301 cells were incubated with vehicle, 10 μM GW6471, or 10 μM GW9662 in the presence of 10 μM cyanidin or 10 μM cyanidin-3G for 24 h. MgCl_2_ (1 mM) was added to the nominally Mg^2+^-free HBSS immediately after the start of the measurements. The increase in fluorescence values for 5 min was compared in each group. (**B**) The HBSS containing 1 mM MgCl_2_ was replaced to the nominally Mg^2+^-free HBSS immediately after the start of the measurements. The decrease in fluorescence values for 5 min was compared in each group. *n* = 4–6. ** *p* < 0.01 and * *p* < 0.05 compared without cyanidin. ^##^
*p* < 0.01 and NS *p* > 0.05 compared with vehicle.

**Figure 8 nutrients-11-00641-f008:**
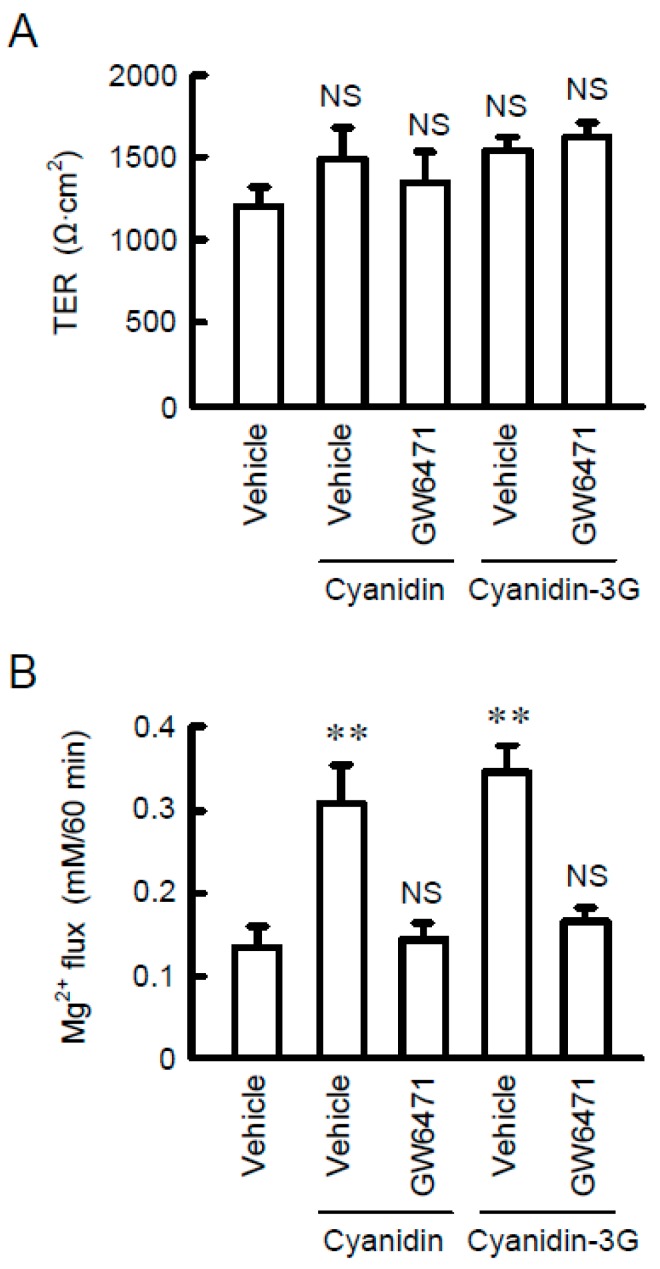
Elevation of transepithelial Mg^2+^ transport by cyanidin. MCE301 cells cultured on Transwell inserts were incubated with vehicle or 10 μM GW6471 in the presence of 10 μM cyanidin or 10 μM cyanidin-3G for 24 h. (**A**) TER was measured using volt ohmmeter. (**B**) The upper and lower chamber solutions were replaced with nominally Mg^2+^-free HBSS. After 1 mM MgCl_2_ was added to the upper chamber solution and incubated for 60 min, the lower chamber solution was collected. The concentration of Mg^2+^ in the lower chamber solution was measured using XB-1 and represented as Mg^2+^ flux. ** *p* < 0.01.

**Figure 9 nutrients-11-00641-f009:**
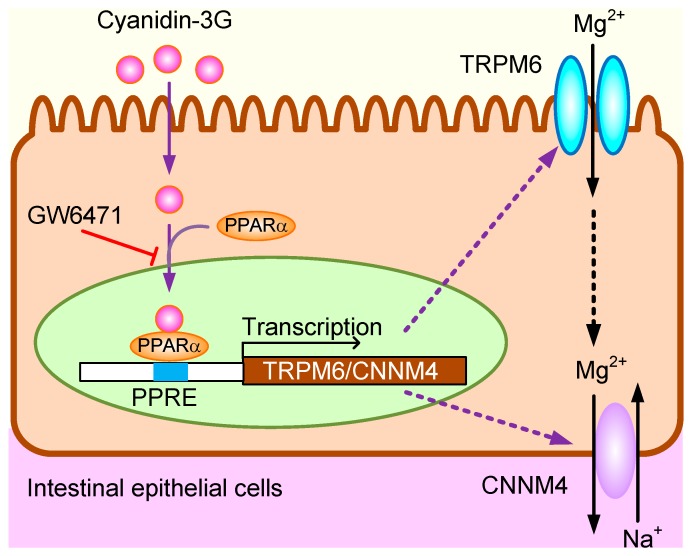
The presumed action mechanism of cyanidine-3G and cyanidin in intestinal epithelial cells.

**Table 1 nutrients-11-00641-t001:** Pharmacological effects of drugs.

Drugs	Mode of Action
GW9662	PPARγ antagonist
GSK3787	PPARδ antagonist
GW6471	PPARα antagonist
H-89	PKA inhibitor

**Table 2 nutrients-11-00641-t002:** Primer list for real time PCR.

TRPM6	Sense	5′-ACCACCTAAGGCAGATGCAA-3′
	Antisense	5′-CAACTTCATTTGGGCTTCTTG-3′
TRPM7	Sense	5′-AACCAACACTCTGGAAGAGATCA-3′
	Antisense	5′-TCAGTCAAGTTTTCTCCCACAC-3′
CNNM4	Sense	5′-TGATGGAGATGTTGAAGGTGAC-3′
	Antisense	5′-CCTCCACAGTTTTGGTCCTTAG-3′
β-Actin	Sense	5′-CCAACCGTGAAAAGATGACC-3′
	Antisense	5′-CCAGAGGCATACAGGGACAG-3′

**Table 3 nutrients-11-00641-t003:** Primer list for the introduction of mutations.

TRPM6	Sense	5′-GACTGAAGGATGCAGTGAGCCATGATCCTGC-3′
	Antisense	5′-CCCAGGCTCAAGTGATCCTTCCACT-3′
CNNM4	Sense	5′-GAATTCTTGCCCCAATTCTCTGGTTAGCAAG-3′
	Antisense	5′-TTACCTCTTACGGCCTTGGTTTCTC-3′

**Table 4 nutrients-11-00641-t004:** Primer list for ChIP assays.

TRPM6	Sense	5′-CCAGGTTTTATGGCTACTGGAC-3′
	Antisense	5′-GACGTATGACTACGGGCTTCTC-3′
CNNM4	Sense	5′-CTGCCATTTTTCTGATGATAGG-3′
	Antisense	5′-TCTGACCTAGGTTTTTCACCTG-3′

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
