# Peer review of "Cyanidin Increases the Expression of Mg2+ Transport Carriers Mediated by the Activation of PPARα in Colonic Epithelial MCE301 Cells"

_nutrients, 2019, doi:10.3390/nu11030641_

Round 1
Reviewer 1 Report
Line 66- May be correcto to black soybean; soybean does not have anthocyanins
Reference 19 in the introduction does not correspond to the sentence
Most of the results were obtained with cyanidin but the biological relevance should be discussed. In nature the main form of cyanidin is cyanidin-3-glucose, so conclusions about the effect of anthocyanins-rich food consumption in Mg transport should be taken with care.
Author Response
We thank you very much for your careful reading of our manuscript and valuable comments.
Following your suggestion, we modified the Discussion. Please see line 344.
Reviewer 2 Report
Dear Editors,
the paper of Takashina et al., described the effects, in vitro, of cyanidin in magnesium intestinal transport (TRPM6 and CNNM4) at transcriptional, translational and functional level. The date is innovative and the message is important to understand the bio-aviability of magnesium and how this is modulated by exogen substances such as cyanidin.
Only few questions are needed:
The TER recording displays no change in resistance, but the influx or efflux of magnesium is expected to impact on resistence. A discussion of this aspect is needed.
Could be helpful to insert a final scheme for a better presentation of the models proposed
Could be helpful to integrate in the method section a table of pharmacological tool used to better read the paper
Author Response
We thank you very much for your careful reading of our manuscript and valuable comments.
Comment 1
The TER recording displays no change in resistance, but the influx or efflux of magnesium is expected to impact on resistence. A discussion of this aspect is needed.
Answer
Following your suggestion, we modified the Discussion. Please see line 335.
Comment 2
Could be helpful to insert a final scheme for a better presentation of the models proposed
Answer
Following your suggestion, we added a predicted scheme. Please see new figure 9.
Comment 3
Could be helpful to integrate in the method section a table of pharmacological tool used to better read the paper.
Answer
Following your suggestion, we added a table of pharmacological tool. Please see line 93.